# Spatialized Life Cycle Assessment of Fluid Milk Production and Consumption in the United States

Andrew D. Henderson [1,2], Anne Asselin-Balençon [1,3], Martin C. Heller [4,5], Jasmina Burek [6,7], Daesoo Kim [6], Lindsay Lessard [8], Manuele Margni [9], Rosie Saad [9], Marty D. Matlock [10], Greg Thoma [6,11], Ying Wang [12,13] and Olivier Jolliet [1,14],*

1. Department of Environmental Health Sciences, School of Public Health, University of Michigan, Ann Arbor, MI 48109, USA
2. Department of Environmental and Occupational Health Science, UTHealth School of Public Health, Austin, TX 78701, USA
3. Sayari, 78112 Saint-Germain-en-Laye, France
4. Center for Sustainable Systems, School for Environment and Sustainability, University of Michigan, Ann Arbor, MI 48109, USA
5. AgResilience Consulting, LLC, Traverse City, MI 49684, USA
6. Department of Chemical Engineering, University of Arkansas, Fayetteville, AR 72701, USA
7. Department of Mechanical Engineering, University of Massachusetts, Lowell, MA 01854, USA
8. Quantis, EPFL Innovation Park—Bât. D, 1015 Lausanne, Switzerland
9. CIRAIG, Department of Mathematical and Industrial Engineering, École Polytechnique de Montréal, Montreal, QC H3T 1J4, Canada
10. Department of Biological and Agricultural Engineering, University of Arkansas, 203 Engineering Hall, Fayetteville, AR 72701, USA
11. AgNext, Colorado State University, Fort Collins, CO 80523, USA
12. Dairy Research Institute, Chicago, IL 60018, USA
13. US Farmers and Ranchers in Action, Chesterfield, MO 63006, USA
14. Quantitative Sustainability Assessment, Department of Environmental and Resource Engineering, Technical University of Denmark, 2800 Kongens Lyngby, Denmark
* Correspondence: ojoll@dtu.dk; Tel.: +1-734-717-6734; Fax: +1-734-936-7283

**Abstract:** Purpose: Understanding the main factors affecting the environmental impacts of milk production and consumption along the value chain is key towards reducing these impacts. This paper aims to present detailed spatialized distributions of impacts associated with milk production and consumption across the United States (U.S.), accounting for locations of both feed and on-farm activities, as well as variations in impact intensity. Using a Life Cycle Analysis (LCA) approach, focus is given to impacts related to (a) water consumption, (b) eutrophication of marine and freshwater, (c) land use, (d) human toxicity and ecotoxicity, and (e) greenhouse gases. Methods: Drawing on data representing regional agricultural practices, feed production is modelled for 50 states and 18 main watersheds and linked to regions of milk production in a spatialized matrix-based approach to yield milk produced at farm gate. Milk processing, distribution, retail, and consumption are then modelled at a national level, accounting for retail and consumer losses. Custom characterization factors are developed for freshwater and marine eutrophication in the U.S. context. Results and discussion: In the overall life cycle, up to 30% of the impact per kg milk consumed is due to milk losses that occur during the retail and consumption phases (i.e., after production), emphasizing the importance of differentiating between farm gate and consumer estimates. Water scarcity is the impact category with the highest spatial variability. Watersheds in the western part of the U.S. are the dominant contributors to the total water consumed, with 80% of water scarcity impacts driven by only 40% of the total milk production. Freshwater eutrophication also has strong spatial variation, with high persistence of emitted phosphorus in Midwest and Great Lakes area, but high freshwater eutrophication impacts associated with extant phosphorus concentration above 100 μg/L in the California, Missouri, and Upper Mississippi water basins. Overall, normalized impacts of fluid milk consumption represent 0.25% to 0.8% of the annual average impact of a person living in the U.S. As milk at farm gate is used for fluid milk and other dairy products, the production of milk at farm gate represents 0.5% to 3% of this annual impact. Dominant contributions to human health impacts are

from fine particulate matter and from climate change, whereas ecosystem impacts of milk are mostly due to land use and water consumption. Conclusion: This study provides a systematic, national perspective on the environmental impacts of milk production and consumption in the United States, showing high spatial variation in inputs, farm practices, and impacts.

**Keywords:** dairy; life cycle assessment; farm; milk production; milk consumption; spatial analysis

## 1. Introduction

Amid concern about the environmental impacts of foods, there is focus on the role of livestock in a sustainable global food system [1–3]. Among livestock systems, milk is of special interest for developing and testing spatialized approaches for assessing environmental impacts: a variety of feed inputs, possibly from disparate geographic areas, are transformed into raw milk, which is then processed and distributed for direct consumption or for use in a variety of products.

A number of Life Cycle Assessments (LCAs) of milk have been conducted; however, they have been limited in the impact categories that were considered, the stages of the dairy consumption life cycle chain (feed production, dairy production, transport, processing, packaging, retail, and consumption) that were included, or the degree to which spatially relevant information was used. Regarding impact categories, most studies have focused on carbon footprint, e.g., in Europe [4,5], Canada [6] or the United States [7–9]. Globally, greenhouse gas estimates for milk at farm gate range from approximately 1.2 kg $CO_2$e/kg Fat and Protein Corrected Milk (FPCM) in North America to over 4 kg $CO_2$e/kg FPCM in portions of Asia and Africa, driven largely by differences in production practices [10]. Capper et al. [11] investigated temporal changes in greenhouse gases from dairy production in the United States (U.S.), showing a 63% reduction in the carbon footprint from 1944 to 2007, with additional reductions in the last decade [12]. Meta-analyses of dairy LCAs, such as an analysis of 13 European milk production studies [13], or an analysis of 44 LCAs of milk production [14], have found that comparability among these studies remains difficult due to variations in functional unit, system boundaries, and transparency; furthermore, they suggest that land use change, biodiversity, water consumption, acidification, and eutrophication warrant further research. Consideration of ecotoxicity or human health impacts (e.g., from respiratory inorganics or toxicity) has been minimal. For example, Cederberg et al. [15] reported total mass of pesticide applied but did not include ecological or human toxicity effects. Few studies have looked across impact categories using normalization; one study that did include normalization identified global warming, acidification, and eutrophication as key contributors to impact [16].

With respect to life cycle stages, most studies evaluate milk production. Fewer studies have considered the milk supply chain past the farm gate: some examples include analyses up to the point of distribution to the consumer in Norway [17], up to the point of retail distribution in Spain [18], and up to the point of disposal by the consumer Sweden [19]. When studies do consider consumption, there is a corresponding reduction in the attention given to milk production. For example, a study of fat spreads (using dairy milk as an input) uses qualitative information to create archetypical mixes of dairy farms by country, evaluating climate, land, and water in spreadable products across countries in Europe and North America [20].

With respect to spatial variation, Yan et al. [13] also emphasized the need for using site-specific emission factors and characterization factors for assessing, e.g., manure management or production of purchased feed. Where spatial differences are modeled, they are typically based on practices on the dairy farm, not differences in feed production (e.g., variability in irrigation), nor differences in characterization (e.g., eutrophication impacts differ based on location of phosphorus or nitrogen emission). Regarding the latter, most

LCA methods, and hence LCA studies, have operated at national inventory levels with global (or possibly continental) characterization of emissions.

There are custom applications of spatially varying characterization factors; for example, Henderson et al. [21] developed a spatialized matrix-based approach for the U.S. that coupled spatially varying rations, feed production, and characterization, focusing on water stress. Guerci et al. [22] compared studies in different regions of Italy, but used standard emission factors for, e.g., ammonia from fertilizer application and it is unclear whether the provenance of purchased feed was considered. In the U.S., a regional analysis of U.S. dairy, accounting for feed provenance, evaluated only climate change impacts [9,23]. Eshel et al. [24] consider climate, land, water, and reactive nitrogen emissions for U.S. dairy, using national-level estimates for production of dairy feed supply. Rotz et al. [25,26] use a regional approach of process modeling to build estimates for farm gate emissions of GHG emissions, energy demand, water consumption, and reactive N loss, but do not fully account for feed variation, instead using a nationally weighted average of irrigated and non-irrigated crops.

While selected areas of concern have been identified in these studies, there is high interest in developing and applying consistent spatialized methods across impact categories to the heterogeneous production system represented by the United States, in which milk production practices, feed production practices, and climate vary. One interesting challenge presented by the dairy system at such a continental level is the number of inputs required to produce dairy and the potential distances between those inputs and the point of milk production. Specifically, milk production often relies on inputs produced locally (e.g., silages) and from afar (e.g., commodity grains). For those environmental impacts with high geospatial variation, it is critical to account for the location of feed production. The product of the dairy farm becomes the input to a national processing and distribution network, such that milk may be consumed thousands of miles from where it was produced. Given consistency in food processing practices, post-farm spatial variation is less critical.

There is therefore the need to assess comprehensive environmental impacts of dairy while capturing variability brought by geographic, climatic, or production practice variation. Regarding the wide distribution and variety of U.S. feed and dairy farms, it is crucial to develop new approaches that are able to reflect the variability in specific production characteristics. We develop such an approach for regions within the U.S., though the approach would be valid whenever a study area encompasses variety in geography, climate, or production practices.

*Objectives*

To address these needs, this paper aims to create a spatially explicit baseline assessment of overall environmental impacts of milk production and consumption in the U.S., identifying hot spots and geographic sensitivities along the production chain and throughout the life cycle and covering both farm gate and consumer levels. More specifically, it aims to:

- Quantify impacts of fluid milk production and consumption, across life cycle stages, focusing on (a) water-related impacts including scarcity-weighted water consumption, (b) eutrophication of marine and freshwater systems, (c) land use, (d) human toxicity and ecotoxicity, (e) climate change;
- Analyze the spatialized distributions of milk production impacts across the U.S., accounting for locations of both feed and on-farm activities, and considering both production quantity and local impact intensity in areas (states or, for water-driven impacts, watersheds);
- Analyze the magnitude of impact associated with overall consumption of fluid milk, compared to overall impacts in the U.S., and analyze tradeoffs between impact categories and between areas.

## 2. Methods

This study builds on the underlying data collected for greenhouse gas studies of U.S. dairy [9,23]; this section further details the major inventory flows and multiple impacts associated with milk production and consumption at a sub-national level.

### 2.1. Functional Unit

The overall functional unit for environmental impacts across the milk life cycle is one kg FPCM consumed in the U.S. (kg FPCM$_{consumed}$). This unit is consistent with the previous carbon footprint upon which this work draws [9,23], and energy-corrected functional units have been shown to be preferable for dairy [27]. For allocation between meat and milk, we applied the biophysical approach of Thoma et al. [23].

While the consumption of milk is an important perspective, an intermediate but equally relevant functional unit is one kg of FPCM produced (kg FPCM$_{farm}$), i.e., up to the point at which milk leaves the farm gate, to enable comparison with most dairy LCAs. Major differences between milk at the farm gate and milk consumed include the allocation to separated cream (19.8%) and losses at retail (12%) and consumer stages (20%) [23,28]. With respect to absolute quantities, a significant fraction of fluid milk at farm gate is used for other dairy products such as cheese and yogurt. Losses of fluid milk at retail and consumer, due to both spoilage and wasting, require that approximately 1.3 kg of fluid milk be produced at the farm gate in order for one kg of milk to be consumed.

### 2.2. Matrix Approach to Spatial Inventory and Impacts

Figure 1 presents the general approach for determining inventory and impacts: Supply chain commodities such as fertilizers and pesticides are modelled at national level. Feed production [$B_{farm}$] is modelled at state level and linked to states of milk production, drawing on rations [$R$] from five milk-producing regions [23] in a matrix-based approach to calculate state impacts, which are then weighted by state milk production [$P$]. (A map of milk-producing regions (Figure S1) and other information are provided in the Supplementary Materials [29–52].) Milk processing, distribution, retail, and consumption are then modelled at national level, accounting for cream allocation and retail and consumer losses. In the spatially explicit portion of the modeling, emissions from an area (state, watershed, or milk-producing region) are connected to spatially explicit impacts in receiving areas (states or watersheds) via the matrix-based characterization factor approach described below.

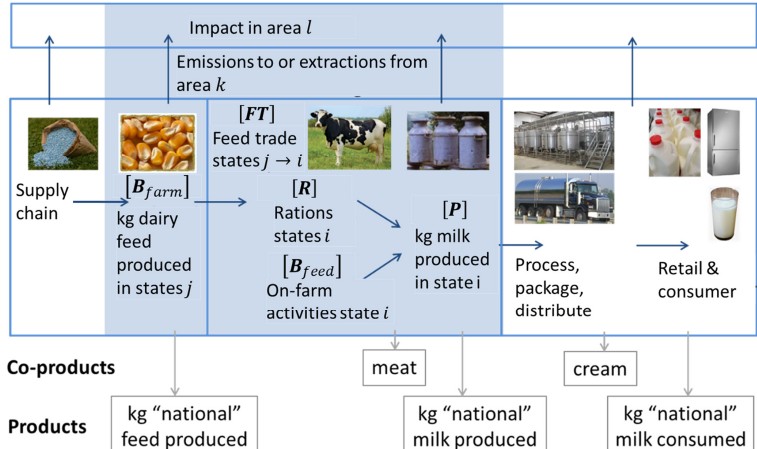

**Figure 1.** Overview of stages, and modeling components of the milk life cycle. Spatialized stages are highlighted in blue. Matrices are denoted with bolded letters in brackets, with a description below. Production of milk in location *i* (here state) requires supply feed production in locations *j*, which induces emission or extraction in locations *k*, which leads to impacts from those emissions or extractions in locations *l*.

### 2.2.1. Overall Impacts at Farm Gate

Analysis of sub-national inventory and impacts for the agricultural (feed and dairy farm) production itself has been structured according to a spatially explicit matrix formulation, which makes it possible to trace impacts due to milk production in one location $i$ (here state) to (a) the location $j$ in which supply feed is produced, (b) the location $k$ in which emission or extraction are taking place, and (c) the location $l$ of impacts from those emissions (Figure 1). Matrices are described in more detail and illustrated by simplified matrices for the water footprint elsewhere [21]. The impacts per kg FPCM$_{farm}$, spatialized for each state ($I^{state}_{milk\ produced}$), are given by Equation (1):

$$I^{state}_{milk\ produced} = CF \times AF_{milk-beef} \times \left[ B_{farm} + \sum_{feed=1}^{n} \left( B_{feed} \times FT_{feed} \times R \right) \right] \qquad (1)$$

where

- $R$ (kg feed $_{consumed}$/kg FPCM$_{farm}$) is the diagonal matrix of the ration for a given feed, expressing the kg feed consumed in each location $i$ per kg milk produced. $R$ is kept as a diagonal matrix rather than a vector to keep information on feed provenance in subsequent calculations.
- $FT_{feed}$ (kg feed $_{produced}$/kg feed $_{consumed}$) is the feed trade matrix, expressing the kg feed produced in location $j$ per kg feed consumed in location $i$.
- $B_{feed}$ (e.g., kg NO$_{3\ to\ water}$/kg feed$_{produced}$) is the inventory matrix for given a feed, expressing the emission or resource extraction in location $k$ from production of 1 kg DM feed in location $j$.
- $B_{farm}$ (e.g., kg NO$_{3\ to\ water}$/kg FPCM$_{farm}$) is the inventory matrix for emission and extractions from dairy cow activities at farm (such as water consumption or pesticide applied to cow housing), expressing the emission or resource extraction in location $k$ from production of 1 kg milk in location $i$.
- $AF_{milk-beef}$ (dimensionless) is the milk-beef allocation factor matrix calculated from the biophysical approach [23].
- $CF$ (e.g., marine eutrophication impact/kg NO$_{3\ to\ water}$) is the characterization factor matrix, expressing the impact in location $l$ per unit inventory flow in location $k$. Because $CF$ is a matrix, the location of both emissions and impacts is tracked.

The major inventory flows and direct impacts related to agriculture and milk production—i.e., cradle to farm gate—were modeled at the state level. For water consumption and freshwater eutrophication, state-based environmental impact results were also apportioned to HUC-2 (two-digit hydrologic unit code) watersheds [53] to best represent impacts on freshwater ecosystems. Supplementary Material Section S1 presents the 18 watersheds considered, as well as the five milk production regions defined according to Thoma et al. [23].

The resulting matrices, at a 50-state resolution, are provided in Supplementary Material Section S10.

### 2.2.2. National Spatial Inventory and Impact of Milk

To determine each state's contribution to the functional unit (kg FPCM$_{consumed}$), which is defined at a national level, we account for state-level production across the U.S., with each state's impact weighted by the quantity of milk production. This approach results in an aggregate representation of national production, designated here as a "national milk". Mathematically, we multiply the impact matrix $I^{state\ or\ watershed}_{milk\ produced}$ by the national production fraction of milk production in each state or watershed ($P^{state\ or\ watershed}_{milk}$), whose diagonal elements represent the fractional contribution of each area to overall production, Equation (2):

$$I^{national}_{milk\ produced} = I^{state\ or\ watershed}_{milk\ produced} \times P^{state\ or\ watershed}_{milk} \qquad (2)$$

where

- $I^{national}_{milk\ produced}$ (e.g., marine eutrophication impact/$kg^{national}_{milk\ produced}$) is the impact of national milk production, with each matrix element expressing impacts received by location $l$ (row) due to milk production in location $i$ (column).
- $P^{state\ or\ watershed}_{milk}$ (unitless) is the production matrix (kept as a diagonal matrix rather than a vector to trace both induced and received impact by location), expressing fraction of national milk production at the level of analysis (state or watershed).

2.2.3. Overall Impacts of Fluid Milk Consumption over Life Cycle Stages

Once the total spatial inventory or impact is calculated from cradle to farm gate, spatial results are combined with national (i.e., not spatially differentiated) contributions both upstream (e.g., due to fertilizer production) and downstream of the farm gate (e.g., due to pulp for packaging, milk processing, transportation and retail). The combined impact, the national-level impact of milk consumption, $I^{national}_{milk\ consumed}$, is calculated as shown in Equation (3). This calculation is performed using an extended version of the regional fluid milk life cycle model originally developed by Thoma et al. and used in spatialized analyses of climate-related impacts of milk production and consumption [9,23,54]. The milk-cream allocation factor ($AF_{milk-cream}$) is applied to the integrated impacts of milk production, processing ($I_{process}$), transportation and distribution ($I_{transport}$), further adding impacts of retail ($I_{retail}$) along with retail food losses ($FL_{retail}$), and impacts at consumer ($I_{consumer}$) before applying the consumer food loss ($FL_{consumer}$) according to the following equation:

$$I^{national}_{milk\ consumed} = \left[ \left\{ \left( \left[ I^{national}_{milk\ produced} + I_{process} + I_{transport} \right] \times AF_{milk-cream} \right) + I_{retail} \right\} \times (1 + FL_{retail}) + I_{consumer} \right] \times (1 + FL_{consumer}) \tag{3}$$

*2.3. Inventory Data and Methods*

This section summarizes the main approaches and data sources used to build each of the above-described matrices.

This study synthesizes data of differing spatial and temporal resolutions from a variety of sources (see Supplementary Material Section S2 for more details). Most farm-specific data, including rations and the prevalence of different manure management systems, were taken from the 2007 survey by Thoma et al. [9], reflecting over 500 U.S. dairy farms. The majority of state-level production data were drawn from the United States Department of Agriculture (USDA) survey and census information, selecting years consistent with the survey (Tables S2 and S3 list these data sources used in this study to complement the survey data). The following paragraph details data use and treatment specific to the present study.

**Feed:** For feed rations, Asselin-Balençon et al. [54] showed that 12 feed rations are able to explain 91% of the variability in the carbon feed print for lactating cows and 98% of the variability in total footprint, when considering 162 feeds used in U.S. across 531 farms. These feeds are alfalfa hay, alfalfa silage, corn grain, corn silage, distiller's dry grains (DDGS) dry, DDGS wet, grass hay, grass pasture, grass silage, soybean, soybean meal, and a feed mix (rations shown in Table S4). The first eleven feeds capture approximately 83% (mass) of the composite national ration, with the feed mix accounting for the balance; the latter was modeled as a mix of 61% corn grain and 39% soybeans based on an analysis of the dominant contributions to the remainder of the ration. Corn grain and soybeans constitute a large fraction of many of the processed feeds (e.g., corn grain is 36% of concentrates) and can be taken as surrogates for many of the crops. The feed mix has non-negligible contributions to the various impact categories, as also identified by other authors [55,56]. As described in Supplementary Materials Section S4, corn grain, soybeans, and processed feeds (DDGS, meal, and feed mix) are modeled as national commodities based on corn grain and soybean shipment data, hays are produced in the milk region (Figure S1) in which they are consumed, and silages and pasture are produced in the state in which they are consumed.

**Nutrient field applications and losses:** Field-level modeling of nutrient applications and losses was performed using the National Nutrient Loss and Soil Carbon (NNLSC)

database [57], based on the EPIC model [58]. In that model, phosphorus losses from agricultural fields are often driven by runoff, so precipitation plays an important role, as does topography. Therefore, phosphorus emissions decrease, e.g., for corn, by approximately a factor of four from east to west, which is consistent with the prevailing trend of decreasing precipitation from the eastern to western U.S. Nitrogen losses are influenced by climate, with warmer climates driving larger releases. Nitrate emissions are largest in the Southeast, with contribution from western areas being approximately half losses from the Southeast, and other areas of the country contributing approximately half again. Ammonia emissions show a similar trend, but with smaller variation.

**Pesticide residues:** For pesticide residues in consumed milk, data from a national milk sampling campaign [59] was used to estimate milk-borne concentrations of pesticides. Of those pesticides detected, the majority are not directly linked to application in the field for dairy cow crops. Rather, they are global, legacy pollutants which are ubiquitous in the environment. Nonetheless, the impacts of these residues were assessed, and their magnitudes were compared to other human health impacts.

**Water consumption:** In keeping with recommendations from the Life Cycle Initiative [60], we include consumption of water withdrawn from water bodies or groundwater reservoirs. In some frameworks, this is called "blue" water, in contrast to "green" water, which is used to distinguish precipitation from extracted water used for irrigation [61].

**Uncertainty:** Uncertainties on inventory flows were systematically assessed using the same type of pedigree approach as in the ecoinvent databases [62], while customizing the approach to each inventory flow, depending on the database from which data are extracted and the corresponding quality of the data used (see Supplementary Material, Section S5).

*2.4. Impact Assessment Methods*

IMPACT 2002+ [63] was selected as the base assessment method, given its widespread use and the fact that it is amenable to modification, a crucial criterion for this study. This method includes assessment capabilities for human toxicity, respiratory (in)organics, ionizing radiation, ozone layer depletion, photochemical oxidation, aquatic/terrestrial ecotoxicity, aquatic acidification, aquatic/terrestrial eutrophication, land occupation, global warming, non-renewable energy, and mineral extraction. IMPACT 2002+ was complemented by more recent indicators from the Life Cycle Initiative for water and climate change and by specific developments to refine the spatial assessment of certain impacts across the U.S., as noted below. Selection of impact assessment method can influence the life cycle analysis; therefore, sensitivity to the impact assessment method was assessed via parallel calculations using ReCiPe [64] and TRACI [65] (see Supplementary Material Section S9). Although there have been updates to these methods, we elected to use previous versions in order to facilitate comparison with other dairy LCAs. The updates to these LCIA methods are expected to have minimal impacts on overall results.

**Climate change impacts:** Based on recommendations from the Life Cycle Initiative [60], in addition to the traditional Global Warming Potentials (GWP-100), we also report the Global Temperature Change Potential for a 100 year time horizon (GTP-100). While GWP reflects absorption of energy, the GTP reflects the temperature change. As also recommended by the Intergovernmental Panel on Climate Change (IPCC) and the Life Cycle Initiative, we use the characterization factors that include the climate-carbon cycle feedbacks [60,66]. Beyond these midpoint impacts, we used the short and long-term damage characterization factors of IMPACT World+ [67] to determine the climate change damages on both human health in disability-adjusted life years (DALYs) and ecosystem quality in potentially disappeared fraction of species, PDF·m$^2$·yr. As we emphasize the difference between GWP and GTP, we have used factors consistent with Impact World+, rather than the latest IPCC factors.

**Water consumption impacts:** Because understanding the influence of spatial variation was a major focus of this study, we include water-related impacts, and we update them with revised normalization factors for this study. Characterization factors relating water

consumption to location-driven water stress were adopted from the AWARE (Available WAter REmaining) approach [68], with U.S. state values from Boulay and Lenoir [69]. We use the default AWARE values, which are marginal characterization factors; marginal factors are appropriate for dairy farms that source feed from a variety of spatially distributed sources. One could apply average characterization factors in areas with high milk production, which would be appropriate for capturing local water consumption (e.g., to produce silages). Given that both approaches can be justified, we selected marginal factors in our spatially explicit analysis in order to avoid de-emphasizing water-stressed regions [70]. Finally, we compare the AWARE water stress impacts to results using the Water Scarcity Index [71], which was used in a spatially explicit water stress analysis of dairy [21].

**Eutrophication and land use:** Freshwater eutrophication impacts for phosphorus were developed using a $0.5° \times 0.5°$ fate factor model [72], coupled with P concentration data from the EPA [73,74] and effect modeling from [75] (Supplementary Materials). Marine eutrophication factors for nitrogen used the state-based fate factors from TRACI [65] and estimated effect data using hypoxia studies from the Mississippi and Chesapeake bays in the U.S. (Supplementary Materials Section S6.3). Geospatially-dependent characterization factors for land use impacts on biodiversity were taken from Pfister et al. [76]. Although more recent factors for land use impacts are available [77,78], the factors from Pfister et al. are at a spatial scale that captures differences at the level of analysis in this study; the Chaudhary et al. factors were coarser than the resolution in this work.

## 3. Results and Discussion

This section first presents a summary of inventory results followed by life cycle impacts of fluid milk for each impact category individually, discussing specific analysis of the spatialized impacts and respective contribution of milk production for climate change, land use, water consumption, freshwater and marine eutrophication, as well as ecosystem and human health. We then compare and normalize damage level impacts across impacts categories, discuss the overall water footprint and compare results also using other impact assessment methods.

### 3.1. Inventory

A summary of main inventory flows is presented in Table 1. Inventory is disaggregated according to stages of the life cycle, separating inventory related to feed production, dairy production (i.e., the dairy farm), and the remainder of the life cycle. For land use and water consumption, the spatialized feed and farm contributions are at least two orders of magnitude larger than the rest of the life cycle. Phosphorus and nitrogen inventory flows from agricultural production are about one order of magnitude greater than the rest of the life cycle, as the contributions to these inventory flows during milk processing are more substantial.

Table 1, section b, presents the final GWP and GTP midpoint values at an aggregated national level. To discuss the effect of the evolution of GWP100 equivalency factors, we present both the results for IPCC values with strong variation in $CH_4$ values: $IPCC_{2007}$ ($GWP_{CH4} = 25$) and $IPCC_{2013}$ GWP100 ($GWP_{biogenic\ CH4} = 34$) with feedback, as recommended by the Life Cycle Initiative [60]. At midpoint level, it is crucial to ensure consistency between the GWP factors used across studies when comparing GWP100 midpoint results: based on the same emissions, using the recommended $IPCC_{2013}$ factors leads to 21% greater kg $CO_2$ eq. at farm gate.

Multiple inventory flows vary substantially within the U.S. For example, water consumption at farm gate can vary by a factor of 20, and P emissions by a factor of 2 around the median value. In comparison, uncertainty on inventory flows is typically of a factor 2 to 3 for most categories (Supplementary Materials Section S5). Full life cycle inventory results are provided in Supplementary Material Section S10.

**Table 1.** Summary of (**a**) major inventory flows per kg FPCM at farm gate and consumed and (**b**) major midpoint footprint indicators for carbon and water scarcity footprints.

| | Unit | per kg FPCM$_{farm}$ (Farm Gate) | | | | per kg FPCM$_{consumed}$ (Consumer) | | | |
|---|---|---|---|---|---|---|---|---|---|
| | | Feed Product. | Milk Product. | Total | State Spatial Variability (10th to 90th %tile) [a] | Feed Product. | Milk Product. | Post Farm Gate | Total |
| **(a) Inventory Flows** | | | | | | | | | |
| Land use | $m^2$ | 1.29 | $1.72 \times 10^{-3}$ | 1.29 | 1.2–2.1 | 1.56 | $2.08 \times 10^{-3}$ | 0.0653 | 1.62 |
| Water consumption | $m^3$ water | 0.174 | $6.66 \times 10^{-3}$ | 0.181 | 0.01–0.39 | 0.21 | $8.04 \times 10^{-3}$ | $6.48 \times 10^{-3}$ | 0.225 |
| Phosphorus (to water) | kg P | $3.28 \times 10^{-4}$ | $3.06 \times 10^{-7}$ | $3.28 \times 10^{-4}$ | $1.6 \times 10^{-4}$–$6.9 \times 10^{-4}$ | $3.95 \times 10^{-4}$ | $3.69 \times 10^{-7}$ | $3.29 \times 10^{-5}$ | $4.28 \times 10^{-4}$ |
| Ammonia (to air) | kg $NH_3$-N | $2.41 \times 10^{-3}$ | $3.07 \times 10^{-3}$ | $5.49 \times 10^{-3}$ | $4.7 \times 10^{-3}$–$7.9 \times 10^{-3}$ | $2.91 \times 10^{-3}$ | $3.71 \times 10^{-3}$ | $5.16 \times 10^{-5}$ | $6.67 \times 10^{-3}$ |
| Nitrate (to water) | kg $NO_3$-N | $1.71 \times 10^{-3}$ | $1.16 \times 10^{-3}$ | $2.87 \times 10^{-3}$ | $2.1 \times 10^{-3}$–$6.4 \times 10^{-3}$ | $2.06 \cdot 10^{-3}$ | $1.39 \times 10^{-3}$ | $9.49 \times 10^{-6}$ | $3.47 \times 10^{-3}$ |
| Nitrogen (other) | kg N | $3.63 \times 10^{-4}$ | $1.20 \times 10^{-4}$ | $4.83 \times 10^{-4}$ | - | $4.37 \times 10^{-4}$ | $1.45 \times 10^{-4}$ | $8.29 \times 10^{-4}$ | $1.41 \times 10^{-3}$ |
| $CO_2$ (fossil) | kg $CO_2$ | 0.221 | 0.0585 | 0.279 | - | 0.266 | 0.0706 | 0.484 | 0.821 |
| $CH_4$ (biogenic) | kg $CH_4$ | $1.46 \times 10^{-6}$ | 0.0278 | 0.0278 | - | $1.77 \times 10^{-6}$ | 0.0335 | $3.00 \times 10^{-4}$ | 0.0338 |
| $N_2O$ | kg $N_2O$ | $3.38 \times 10^{-4}$ | $4.11 \times 10^{-4}$ | $7.49 \times 10^{-4}$ | - | $4.08 \times 10^{-4}$ | $4.95 \times 10^{-4}$ | $2.28 \times 10^{-5}$ | $9.26 \times 10^{-4}$ |
| **(b) Midpoint indicators** | | | | | | | | | |
| Water consumption—AWARE | $m^3$ world avg ($m^3$ max. deprivation) | 8.4 (0.084) | 0.15 (0.0015) | 8.55 (0.0855) | 0.01–18 (0.0001–0.18) | 10.1 (0.101) | 0.18 ($1.8 \cdot 10^{-3}$) | 0.215 ($2.15 \times 10^{-3}$) | 10.5 (0.105) |
| Water consumption—WSI | $m^3$ in competition | 0.118 | $2.71 \times 10^{-3}$ | 0.121 | $5.0 \times 10^{-4}$–0.25 | 0.143 | $3.27 \times 10^{-3}$ | $2.96 \times 10^{-4}$ | 0.146 |
| Global warming GWP100 shorter term (IPCC 2007) | kg$CO_2$ eq. shorter term | 0.332 | 0.879 | 1.21 | - | 0.401 | 1.06 | 0.582 | 2.04 |
| Global warming GWP100 shorter term (IPCC 2013) | kg$CO_2$ eq. shorter term | 0.337 | 1.13 | 1.47 | - | 0.406 | 1.36 | 0.602 | 2.37 |
| Global warming—GTP100 long term (IPCC 2013) | kg$CO_2$ eq. long term | 0.327 | 0.488 | 0.815 | - | 0.394 | 0.588 | 0.526 | 1.51 |

[a] Dashes indicate impacts that were calculated at a national level, with no variation among states.

## 3.2. Greenhouse Gas Impacts of Milk Production and Fluid Milk Consumption

Figure 2 presents the overall GHG impacts from four perspectives: GWP vs. GTP, showing either emission flows or the life cycle stages where induced emissions occur. We first analyze the contribution of each life cycle stage, differentiated by emission flow type. Agriculture production up to farm gate dominates the GWP100 impacts (78% of overall life cycle impacts), due to emissions of $CO_2$ and $N_2O$ during feed production and substantial contributions of $CH_4$, $N_2O$ and $CO_2$ at dairy farm (Figure 2a). Methane ($CH_4$) accounts for 65% of total GWP impacts. Refrigerant associated emissions ('Other') also play a small but non-negligible role in the transport and retail stages. In the framework of longer-term impacts as characterized by the GTP100 (Figure 2b), the contribution of methane is reduced by 38% across the life cycle relative to GWP. This change results from the $CH_4$ GTP100 factor of 11, which is three times lower than the GWP100 of 34 for biogenic methane. The change in the methane factor reflects the fact that many of the impacts of methane take place before the 100-year time horizon, due to its shorter half-life in the atmosphere.

In the overall life cycle impact, up to 30% of the milk produced is lost during retail and consumption phases, implying that impacts (per kg FPCM) for consumption are about 30% greater than for production. Since field and on-farm impacts are proportionally high, most of the impacts associated with losses at retail/consumption occurs prior to the farm gate, and these losses are generally represented as part of the impact associated with milk production. To reflect that these losses are induced by the retail and consumption stages of the life cycle and the related practices during these stages, Figure 2c,d show the *induced* impacts of each life cycle stage, with losses assigned to the retail and consumer stages, while disaggregating these losses according to other life cycle stages or flows. Figure 2c,d clearly illustrate the substantial contribution of food losses per kg milk consumed. These panels also explicitly show the allocation to cream (close to 20% of the pre-farm gate and processing & transport impacts), which is included but not explicitly shown in the upper panels.

The same factors related to retail losses, consumer losses and cream allocation apply equally to all other impact categories, with retail and consumer impacts responsible for an important share of the induced impacts. To avoid duplication, subsequent graphs only present results for other categories by constituent flows, with loss-induced impacts occurring at the stage of emission.

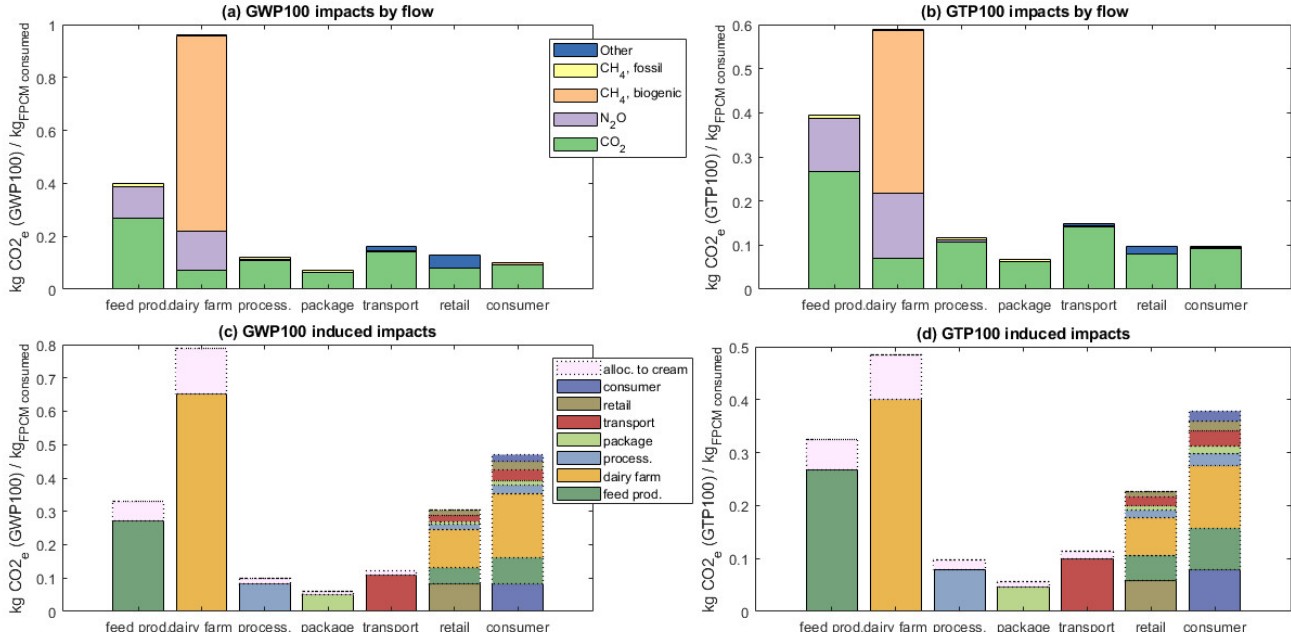

**Figure 2.** Greenhouse gas impacts across the milk consumption life-cycle due to (**a**) GWP100, with flows shown, (**b**) GTP100 with flows shown, (**c**) GWP100 with induced impacts across life cycle stages shown, and (**d**) GTP100 with induced impacts across life cycle stages shown. In all figures, the retail and consumer losses induce additional production of milk, and thus impacts, at earlier life cycle stages. In panels (**a**,**b**), these losses are as induced in the stages where they occur (e.g., additional milk production at farm gate is required for losses at consumer); in (**c**,**d**), they are explicitly shown where the losses occur, as dashed components of retail and consumer stages. For panels (**c**,**d**), both life cycle stages and cream allocation are explicitly shown.

It is also important to differentiate between farm gate and consumer impact estimates, the latter being 55% greater due to losses and processing emissions that occur post-farm gate. This emphasizes the need in all LCA studies to (a) systematically report the main greenhouse gas emission flows ($CO_2$, $CH_4$, $N_2O$, and refrigerant contributors) so that global warming impacts can be easily recalculated and checked and (b) to have all studies reporting at least the $IPCC_{2013}$ GWP100 with carbon cycles as recommended by the Life Cycle Initiative [60].

As a national U.S. average, the greenhouse gas midpoint value per kg milk at farm gate is 1.21 kg $CO_2$ eq. shorter term/kg FPCM for $GWP100_{2007}$, 1.47 kg $CO_2$ eq. shorter term/kg FPCM for $GWP100_{2013}$ and 0.815 kg $CO_2$ eq. long term/kg FPCM for GTP100. All values are in the same range as those obtained with $GWP100_{2007}$ by Thoma et al. [23] and by Baldini et al. [14], in their review of 73 papers on milk production, with 25th and 75th percentiles impacts of 1 and 1.25 kg $CO_2$ eq. shorter term/kg FPCM, and by Fantin et al. [16], with an average value of 1.11 kg $CO_2$ eq/kg FPCM and a standard deviation of 0.23 kg $CO_2$ eq.

### 3.3. Water Consumption Impacts

**Water consumption:** Water consumption is driven by irrigation during feed production (96%), with very minor contributions from the dairy farm, milk processing, or consumer (Figure 3a). Water consumption (as well as scarcity) is the impact category with the highest spatial variability. Figure 4a presents the spatial distribution of water consumption flows across U.S. watersheds, representing the fraction of the national milk production at farm gate, as produced in each main watersheds on the x-axis, and the amount of water consumed per kg $FPCM_{farm}$ in each watershed on the y-axis. As a result, the area associated with each watershed represents the contribution from this watershed to the national water consumption. The sum of the areas represents the total consumption per kg milk of national production, and the cumulative water consumption is shown by the line (secondary y-axis).

Western milk-producing watersheds such as the Lower Colorado (watershed number (#)15, about 0.6 m$^3_{water}$/kg FPCM$_{farm}$), the California (#18, more than 0.4 m$^3_{water}$/kg FPCM$_{farm}$), the Pacific Northwest (#17), and the Rio Grande (#13) are the dominant contributors to the total water consumed at the national level, with 80% of the water consumption associated with only 40% of the total milk production (Figure 4a, cumulative line on secondary right y-axis). Watersheds in areas with lower irrigation requirements (less than 0.01 m$^3_{water}$/kg FPCM$_{farm}$), such as the Upper and Lower Mississippi (#7,8), the Great Lakes (#4) or the Mid-Atlantic (#2) contribute approximately 45% of national milk production but account for less than 1% of water consumption.

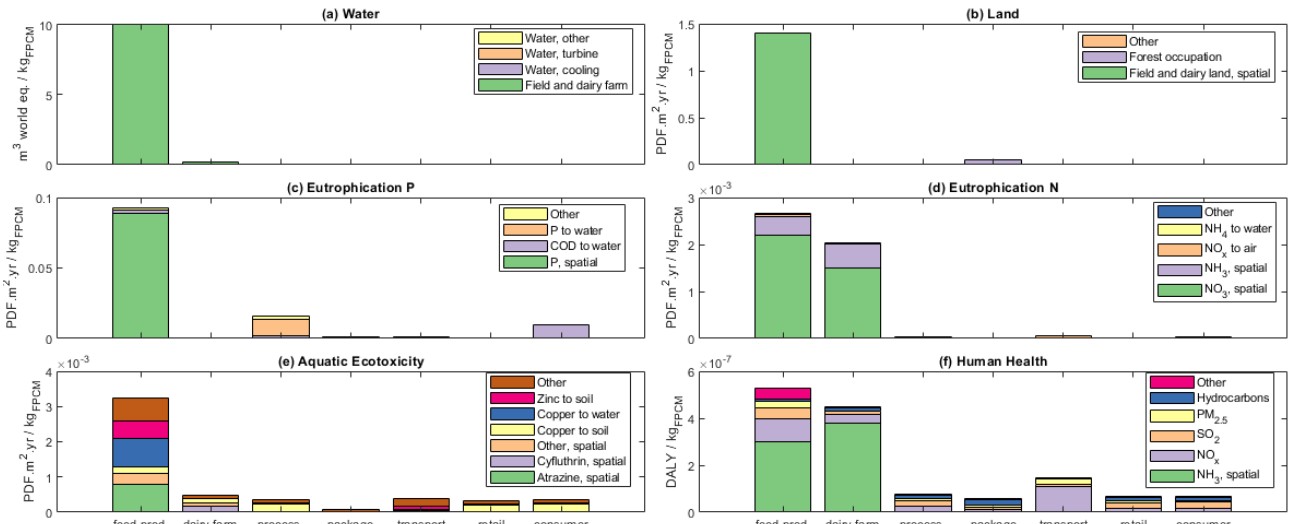

**Figure 3.** Overall fluid milk life cycle impacts per kg FPCM$_{consumed}$ for (**a**) water scarcity (as m$^3$ world equivalents) (**b**) land use (as potentially disappeared fraction of species, PDF, covering 1 m$^2$ over 1 year), (**c**) freshwater eutrophication from phosphorus, P (as PDF·m$^2$·yr), (**d**) marine eutrophication from nitrogen, N (as PDF·m$^2$·yr), (**e**) aquatic ecotoxicity (as PDF·m$^2$·yr), (**f**) overall human health (as disability-adjusted life year, DALY). Induced impacts due to losses at consumer and retail are shown at those life cycle stages (e.g., additional milk production at farm gate is required for losses at consumer).

At a national U.S. average, the water consumption per kg milk amounts to 0.174 m$^3_{water}$/kg FPCM$_{farm}$ and to 0.225 m$^3_{water}$/kg FPCM$_{consumed}$ (the latter accounts for losses, as shown in Table 1a). Based on estimates for water use in the United States by source, type, and sector [79], we find a total US annual freshwater use of 2.85 × 10$^{11}$ m$^3$ (excluding thermoelectric cooling) and total farm use (irrigation and livestock) of 1.8 × 10$^{11}$ m$^3$. Thus, milk production represents 8.2% of the U.S. agriculture water consumption, and 5.1% of the total U.S. water consumption. Fluid milk consumption represents only 1.4% of the total U.S. water consumption, since a large fraction of the milk is used for other dairy products.

**Water scarcity index:** When accounting for the AWARE water scarcity index (Figure 4b), the distribution of water impacts is further skewed towards the driest watersheds such as lower Colorado (#14) and California (#18), which have some of the larger AWARE characterization factors in the U.S., with 90% of the water scarcity impacts associated to only 40% of the national milk production. When considering feed types, water scarcity is the only category with dominant impact from alfalfa hay, corn silage and the feed mix. Overall, the national water scarcity assessment of milk production is dominated by those areas with a combination of high water stress and high milk production, which is driven by local production of hay and silage. AWARE being primarily a comparative index, the absolute value of 8.55 m$^3$/kg FPCM$_{farm}$ has limited meaning. Normalizing by the AWARE maximum deprivation potential of 100 m$^3$ world equivalent [68] enables us to interpret the water scarcity footprint in term of m$^3$ max. deprivation equivalents, with

values of 0.0855 $m^3_{max.deprivation}$ per kg FPCM$_{farm}$ as a U.S. national average (Table 1b). In comparison, the water stress index (WSI, [71]) leads to similar results, with 0.12 $m^3$ water in competition per kg FPCM$_{farm}$ (Table 1b) as discussed by Henderson et al. [21].

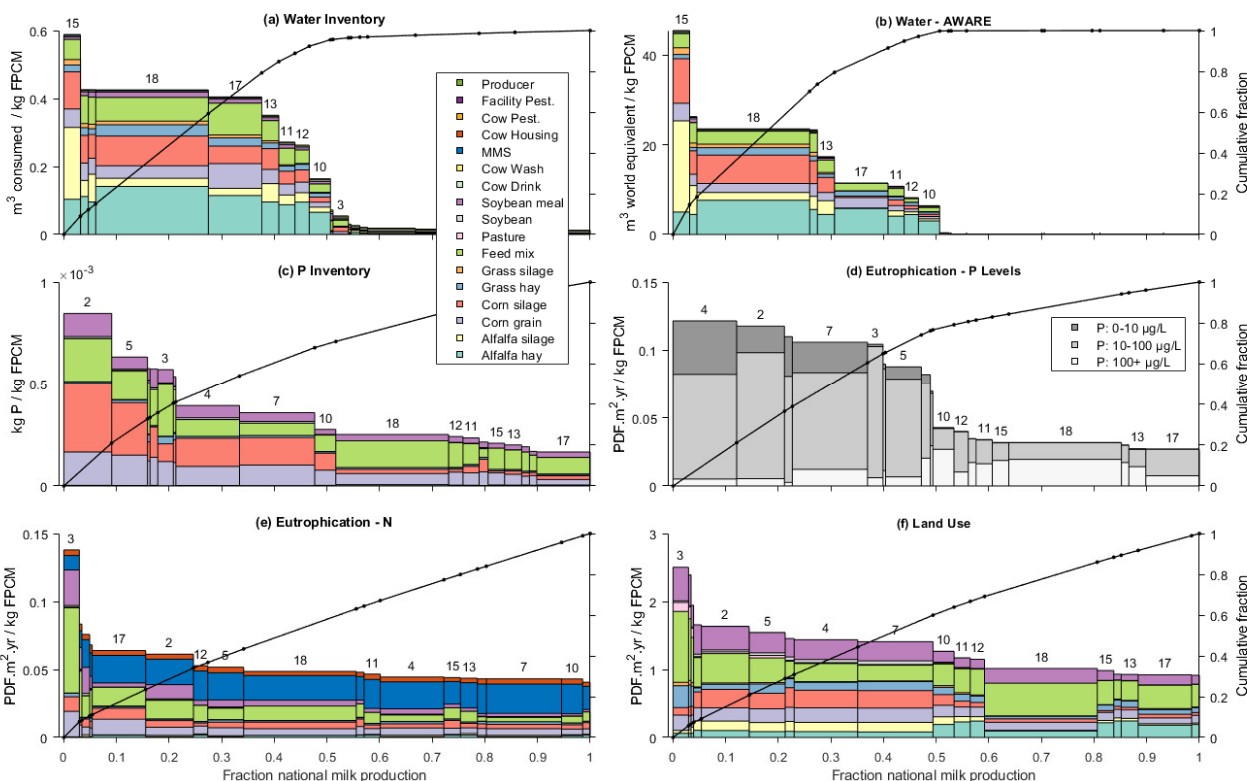

**Figure 4.** Fluid milk inventory and impacts per kg FPCM at farm gate at the watershed level for (**a**) water consumption ($m^3$ of water consumed), (**b**) water scarcity AWARE values (as $m^3$ world equivalents) (**c**) phosphorus (P) emissions to freshwater, (**d**) freshwater eutrophication impacts from P (as potentially disappeared fraction of species, PDF, covering 1 $m^2$ over 1 year), (**e**) marine eutrophication impacts (as PDF·$m^2$·yr), and (**f**) land use impacts (as PDF·$m^2$·yr). Impacts per kg FPCM in each watershed (HUC-2 numbers; see Table S1 for watershed names) are plotted on the y axis, sorted from highest to lowest, and the watershed's fraction of national milk production on the x axis; bar areas thus represent contribution to national-level impacts. Cumulative impact is shown as a line on the secondary y axis. Legend showing feed and farm activities applies to all panels except (**d**), which differentiates impacts by P levels in receiving watersheds. Only watersheds above thresholds for production (x-axis) and impact (y-axis) are labelled.

## 3.4. Freshwater and Marine Eutrophication

**Freshwater eutrophication:** Freshwater eutrophication is primarily associated with phosphorus losses due to soil erosion in feed production areas, since phosphorus (P) is the limiting nutrient in most freshwater bodies. However, P releases during milk processing and the chemical oxygen demand (COD) associated with milk losses at the consumer also represent non-negligible contributions (Figure 3c). P emissions at the milk processing plant are approximately 1/10th of the total feed production impact, provided that appropriate phosphorus reduction technologies are used. A sensitivity study showed that these impacts at processing plants may be up to a factor 7 greater if processing effluent is not adequately treated [80].

Figure 4c presents the phosphorus emissions per kg milk produced in each watershed. The largest emissions occur in regions with high erosion rates associated with field production on slopes, in particular the Mid-Atlantic watershed (#2), which is located in the

foothills of the Appalachian Mountains. In terms of feed, corn grain, corn silage and feed mix are the dominant contributors to these P emissions.

Figure 4d shows the overall impacts at the farm gate for freshwater eutrophication. Impacts are disaggregated according to the phosphorus concentration of receiving water bodies, which allows impacts in relatively pristine waters (e.g., water bodies with P < 100 μg/L) to be included or excluded easily. Impacts in water bodies with existing total P concentrations lower than 100 μg/L account for the majority of impacts. The Great Lakes watershed (#4) has high impacts because of the high phosphorus residence time, and thus highest characterization factors. In contrast, many areas, including the high milk production western watersheds have low freshwater residence time, which means that the phosphorus fate factor is low as well. However, when restricting impacts to areas with existing high levels of eutrophication—i.e., phosphorus concentration greater than 100 μg/L (the lower, pale gray section in Figure 4d)—the highest impacts are induced in the California (#18), Missouri (#10), and Upper Mississippi (#7) water basins where the highest P concentrations are observed. The overall U.S. national average of P emissions per kg milk consumed is $4.3 \times 10^{-4}$ kg P/kg $FPCM_{consumed}$, which is at the low end of the range reported by Baldini et al. [14]: $4 \times 10^{-4}$ kg P/kg $FPCM_{consumed}$ (average estimates of $1.3 \times 10^{-3}$ kg P/kg $FPCM_{consumed}$). Note that Baldini et al. utilize the CML 2001 impact assessment method, which also accounts for N-based eutrophication in "eutrophication potential," but which is considered here separately in the marine eutrophication category. In our model, the COD flow, due primarily to losses during milk processing and modeled at a national level, represents 7.5% of the total freshwater eutrophication impact. The overall freshwater eutrophication impact is 0.15 PDF·m²·yr/kg $FPCM_{farm}$ and 0.21 PDF·m²·yr/kg $FPCM_{consumed}$ (Table 2).

**Table 2.** Summary of impacts, at damage level, per kg $FPCM_{farm}$ and per kg $FPCM_{consumed}$.

| Area of Protection | Impact Category | Unit | Impact per kg $FPCM_{farm}$ | Spatially-Modeled State Variability (10th–90th Percentile)[a] | Impact per kg $FPCM_{consumed}$ |
|---|---|---|---|---|---|
| Human health | Carcinogens | μDALY[b] | $2.1 \times 10^{-2}$ | $8.0 \times 10^{-4}$–$1.8 \times 10^{-3}$ | $9.6 \times 10^{-2}$ |
| Human health | Non-carcinogens | μDALY | $4.1 \times 10^{-2}$ | (combined with carcinogens) | $7.3 \times 10^{-2}$ |
| Human health | Fine particulate | μDALY | 0.75 | - | 1.2 |
| Human health | Ionizing radiation | μDALY | $7.1 \times 10^{-4}$ | - | $2.9 \times 10^{-3}$ |
| Human health | Ozone layer | μDALY | $3.1 \times 10^{-5}$ | - | $7.4 \times 10^{-4}$ |
| Human health | Photochemical oxidant formation | DALY | $2.8 \times 10^{-3}$ | - | $3.7 \times 10^{-3}$ |
| Human health | Water consumption | μDALY | $5.7 \times 10^{-4}$ | $2.9 \times 10^{-6}$–$1.5 \times 10^{-3}$ | $6.9 \times 10^{-4}$ |
| Human health | Climate change shorter term (0–100 yr) | μDALY | 1.2 | - | 1.9 |
| Human health | Climate change long term (>100 yr) | μDALY | 1.0 | - | 2.1 |
| **Human health** | **Total** | **μDALY** | 3.02 (0.69%)[c] | - | 5.45 (0.26%)[c] |
| Ecosystem Quality | Climate change shorter term (0–100 yr) | PDF·m²·yr[d] | 0.26 | - | 0.42 |
| Ecosystem Quality | Climate change long term (>100 yr) | PDF·m²·yr | 0.23 | - | 0.49 |
| Ecosystem Quality | Land use biodiversity | PDF·m²·yr | 1.1 | 0.87–2.3 | 1.4 |
| Ecosystem Quality | Terrestrial acidification/nutrification | PDF·m²·yr | 0.11 | - | 0.15 |
| Ecosystem Quality | Aquatic ecotoxicity | PDF·m²·yr | $3.1 \times 10^{-3}$ | $2.3 \times 10^{-4}$–$7.1 \times 10^{-4}$ | $7.0 \times 10^{-3}$ |
| Ecosystem Quality | Freshwater eutrophication (Total) | PDF·m²·yr | 0.15 | 0.02–0.11 | 0.21 |
| Ecosystem Quality | Marine eutrophication | PDF·m²·yr | $4.7 \times 10^{-2}$ | $2.7 \times 10^{-3}$–$7.6 \times 10^{-3}$ | $5.8 \times 10^{-2}$ |
| Ecosystem Quality | Water consumption | PDF·m²·yr | 0.40 | $1.6 \times 10^{-3}$–0.32 | 0.50 |
| **Ecosystem Quality** | **Total** | **PDF·m²·yr** | 2.3 (2.9%)[c] | - | 3.2 (0.8%)[c] |
| Resources | Water consumption | MJ primary[e] | 0.59 | - | 0.71 |
| Resources | Non-renewable energy | MJ primary | 4.4 | - | 15 |
| Resources | Mineral extraction | MJ primary | $2.0 \times 10^{-3}$ | - | $5.1 \times 10^{-3}$ |
| **Resources** | **Total** | **MJ primary** | 5.0 (0.48%)[c] | - | 15.7 (0.31%)[c] |

[a] Dashes indicate impacts that were calculated at a national level, with no variation among states. National level impacts (per kg $FPCM_{farm}$ and per kg $FPCM_{consumed}$) can exceed state variability because the state variability reflects only the portion of the life cycle that was modeled spatially and because states with high milk production may be outside the reported percentiles. [b] A μDALY or $10^{-6}$ DALY correspond to 1 per million of a disability-adjusted life year, or DALY. Since there are 31.5 million seconds in a year, 1 μDALY corresponds to 31.6 s or 0.53 min of healthy life lost [81]. [c] Normalized total impacts in parenthesis. [d] A potentially disappeared fraction of species, PDF, covering 1 m² over 1 year. [e] Mega-joule.

**Marine eutrophication:** Marine eutrophication is primarily associated with fertilizers and related N emissions, nitrogen typically being the limiting nutrient in marine environments, such as the Gulf of Mexico or the Chesapeake Bay. In contrast to phosphorus emissions, which are associated with crop production, a large fraction of nitrogen emissions, and thus impacts, are related to the dairy farm itself, through ammonia emissions from dairy cow housing and manure storage (Figures 3d and 4e). Figure 4e shows overall national spatialized impact due to both ammonia and nitrate emissions. Because of smaller variations in feed production and dairy farm emissions, and smaller variations in characterization factors, the variation in marine eutrophication impact between watersheds is limited. Of these two primary nitrogen species, impacts due to nitrate outweigh impacts due to ammonia at a ratio of approximately 3 to 1, with on-farm $NH_3$ contributing to 23% of marine eutrophication impacts (Figure 3d). This relationship stems from several factors. First, nitrate has slightly higher characterization factors in the TRACI model [65], because the fraction of nitrogen delivered to marine systems is greater for nitrate emissions to water than for ammonia emitted to air. Secondly, the EPIC model predicts higher nitrate emissions to water than ammonia emissions to water. Finally, the IPCC estimate of nitrate loss from manure management systems is a constant fraction of 10% across all types of manure management systems. Beyond ammonia and nitrate, nitrogen emissions to water and nitrogen oxide emissions to air account for 0.5% and 3.2% of the overall impact, respectively. The overall marine eutrophication impact is 0.047 PDF·m$^2$·yr/kg FPCM$_{farm}$ and 0.058 PDF·m$^2$·yr/kg FPCM$_{consumed}$ (Table 2).

*3.5. Land Use*

Land use impacts for milk production are driven by feed production, while impacts due to occupation by dairy farms are negligible (Figure 3b). Connecting feed to land use via the yield and feed distribution has allowed this study to create a more nuanced picture of land use for milk production than would be achieved via a purely non-spatial assessment. Figure 3b also indicates that the land for pulp production for paperboard and other packaging are a limited, but non-negligible contributor to overall impact.

Figure 4f presents the watershed-national comparison of biodiversity impacts due to land use from field and dairy farm. At the watershed level, the variations in land use impacts are restricted in comparison to variations in categories such as water consumption, the only greater impacts occurring in the South Atlantic-Gulf (#3), in particular due to lower milk productivity per cow. Western watersheds (#17 and #18) have slightly lower land use inventory requirements than other watersheds, due to a lower fraction of silage in the rations. However, the main driver for these small differences between watersheds in Figure 4f is a higher characterization factor for land use biodiversity impacts in the East and Midwest, as opposed to the West. In addition, the South Atlantic-Gulf watershed (#3) tends to have moderately lower regional yield averages than other areas, leading to a greater land use requirement.

As a national U.S. average, the land use requirement is 1.3 m$^2$·yr/kg FPCM$_{farm}$. The 2007 USDA Census of Agriculture [82] provides estimates of farmland (922.1 M acres, or 373.1 M hectares, ha) and cropland (406.4 M acres, or 164.5 M ha). Nickerson et al. [83] estimates total agricultural land (including forestry) at 1159 M acres (469 M ha), cropland at 408 M acres (165 M ha), and total land in the lower 48 states at 1894 M acres (766.5 M ha). We use the total US land value of Nickerson and the farmland estimate from the Census, as the latter captures both livestock and crops, but excludes forestry. Thus, the land use requirement for milk production represents 2.2% of the U.S. agriculture land use and 1.4% of total U.S. land use. The corresponding land use biodiversity impacts are 1.1 PDF·m$^2$ ·yr/kg FPCM$_{farm}$ and 1.4 PDF·m$^2$·yr/kg FPCM$_{consumed}$.

*3.6. Ecotoxicity and Human Health Impacts*

**Aquatic ecotoxicity:** Figure 3e shows the ecotoxicity impacts across the milk production life cycle. Spatially differentiated impacts due to atrazine, metolachlor, and other

pesticides account for close to half of the feed production impact. Cyfluthrin and other pesticides applied to cows or to cow facilities account for the other half of the dairy farm impact. The remainder of the ecotoxicological impacts associated with feed and milk production are due to emissions of metals, stemming from the nonspatial life cycle inventory.

**Respiratory organics and human toxicity:** Figure 3f shows the human health impacts across the milk life cycle. The majority of impacts are caused by emissions of ammonia, nitrogen oxides, and sulfur dioxides and the subsequent formation of secondary fine particulate smaller than 2.5 μm. These substances are important across all stages of the fluid milk life cycle, with ammonia emissions largely due to feed production and manure management, $NO_x$ from tractor operations and milk transport, and other substances associated primarily with fuel use. The human toxicity impacts of pesticides and other substances were found to be an order of magnitude smaller than human health impacts due to respiratory inorganics. Combined human health impacts are on the order of $1 \times 10^{-6}$ DALY or 1 μDALY/kg $FPCM_{consumed}$, mostly due to ammonia.

*3.7. Damage Impacts across Impact Categories and Normalized Results*

Table 2 summarizes impact results, grouped according to areas of protection (human health, ecosystem quality, resources, and climate change). At the damage level, human health impacts are measured in DALYs, ecosystem quality is expressed in terms of PDF·m$^2$·yr; resources are assessed as equivalents quantity of megajoules of primary, non-renewable, energy (MJ primary); For climate change, we report separately the damage according to Impact World+ short term (<100 years) and long-term (>100 years). Results are presented for the two functional units considered: impacts per kg $FPCM_{farm}$, and per kg $FPCM_{consumed}$. Differences between these two values are driven by allocation (e.g., between cream and liquid milk), losses at the retail and consumer stages, as well as other inputs at the post-farm life cycle stages (e.g., energy for transport and refrigeration).

Values presented in Table 2 are not directly comparable across areas of protection. However, it is possible to relate them to a common point of reference: the total damage level impact generated by one person, accounting for all impact categories within a given area of protection. This process is called normalization and is presented for impacts per kg FPCMconsumed (Figure 5), using the normalization values of the modified IMPACT2002+ method developed specifically for the U.S. population (Table S15). For production, values are normalized considering the total annual production of U.S. fluid milk per capita of 276 kg FPCM produced/pers/yr (see Figure S7). Because produced fluid milk is used for a variety of dairy products, the actual consumption of fluid milk is lower, and we use an average value of 58 kg FPCM consumed/pers/yr for the normalization of consumed milk (Figure 5). In each category, normalized damages may be interpreted as the fraction of total lifestyle impact related to milk production (or consumption) and damages can be compared within each Area of Protection. Comparing across areas of protection is not recommended, since it would implicitly assume equal weighting across these areas (i.e., across impact per person on human health, ecosystem quality, and resources). Alternative weighting factors for damages, such as the Stepwise factors (Table S15) could provide insights on the respective magnitude of these impacts. Given the uncertainty of inventory data, impact assessment methods, and damage assessment level, this comparison primarily enables us to distinguish the impact categories substantially contributing to damages within each area of protection from those smaller effects (generally one to three orders of magnitude smaller than dominant impact categories).

**Human health:** Impact of fine particulate (respiratory inorganics in figure) represent one of the most important contributions to the life cycle impacts of milk production on human health (resp. inorganics, Figure 5). These impacts are driven by (a) on farm emission during milk production (31% due to on-site $NH_3$ emitted from barn and from manure management systems), (b) feed production (39% mainly due to ammonia from manure and synthetic fertilizer, as well as NOx and $PM_{2.5}$ impacts of tractors), and (c) to a lesser extent, milk transportation (10% also mainly due to NOx and PM2.5 emissions).

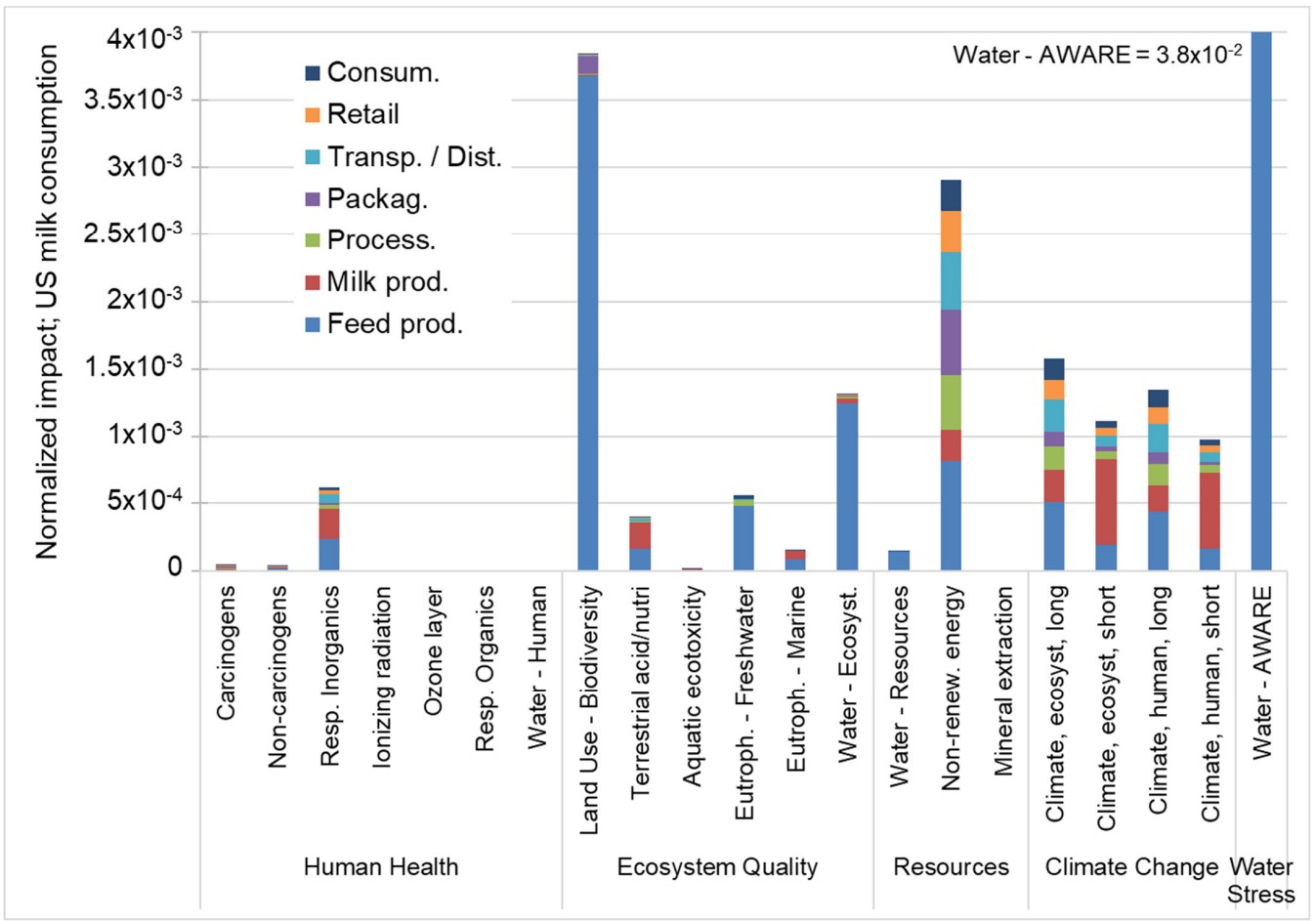

**Figure 5.** Normalized impacts for the annual consumption of 58 kg of U.S. fluid milk per person (see Figure S6 for logarithmic scale).

Carcinogenic impacts are an order of magnitude lower than respiratory inorganics and are mainly due to indirect electricity production processes along the entire life cycle. Non-carcinogenic impacts are of the same order of magnitude as carcinogenic impacts and occur mainly during feed production, associated in particular with the metals content of phosphorus fertilizer. Dieldrin, a legacy pesticide residue, contributes to less than 1% to the overall non-cancer human health impact, with the other pesticides in milk [59] having negligible contributions on a national basis. Indirect impacts of pesticides (mainly atrazine) applied to feed crops also have limited contributions that are over a factor of 10 smaller than dominant carcinogenic impacts and a factor of 1000 smaller than the dominant non-carcinogenic impacts.

The other impact categories (ionizing radiation, ozone layer depletion, photochemical oxidant formation) only play a minor role in human health impacts, being 3 to 5 orders of magnitude below the normalized impacts of fine particulate.

Using Impact World+ [67] to estimate the order of magnitude of climate change impacts on human health suggests that this impact category has a dominant contribution to human health impacts, with 40% of the impact happening in the first 100 years, mostly due to methane emissions from milk production, plus another 60% of longer-term impacts associated with $N_2O$ and $CO_2$ from feed production and all other life cycle stages. This difference is driven by the fact that 100 years is sufficient to consider most of the impact of $CH_4$, as it has a limited atmospheric lifetime, but that a large share of $CO_2$ and $N_2O$ effects still occur after 100 years. The climate change impacts per kg FPCM$_{consumed}$ represent approximately 0.5% of the total normalized impacts of a U.S. person.

Human health impacts of fluid milk consumption represent 0.26% of the annual average impact of a person living in the U.S. (Figure 5), 5.5 μDALY/kg $FPCM_{consumed}$, with 1.2 μDALY/kg $FPCM_{consumed}$ for respiratory inorganics impact and 4.1 μDALY for climate change impacts. Interestingly, this is in the same order of magnitude as the nutritional beneficial effects of milk on reducing colon cancer, which is on the order of 4.5 μDALY per kg FPCM [81,84]. The overall impacts on human health of raw milk production (i.e., at farm gate) represent 0.7% of the annual average impact of a person living in the U.S.

**Ecosystem quality:** Several impact categories contribute to reduced ecosystem quality. Land use impacts on terrestrial biodiversity have the highest contribution of all normalized impacts, mainly due to feed production, with a limited contribution of paper production for packaging. Climate change on ecosystems shows as the second main contributor to impacts on ecosystem. Since crop irrigation is the major driver for water consumed towards milk production, water consumption impacts on biodiversity are entirely dominated by feed production. Acidifying and eutrophying emissions to air also contribute to terrestrial impact, driven by on-farm emissions of $NH_3$ to air from dairy cow barns and manure handling and storage. For impacts on freshwater ecosystems, eutrophication due to phosphorus emissions during feed production represents an important contribution. Aquatic ecotoxicological impacts of pesticides (mainly atrazine) and of heavy metals in phosphate fertilizers are relatively small in comparison with other normalized impacts on ecosystems.

Overall impacts on ecosystem quality of fluid milk consumption represent on the order of 0.8% of the annual average impact of a person living in the U.S. (Table 2), whereas milk production represents 2.3% of the ecosystem impacts per U.S. person per year, mostly due to land use, water consumption, climate change and eutrophication.

**Resources and non-renewable energy:** Direct uses of energy across the life cycle dominate non-renewable energy use; these include fuel for tractors and transportation and electricity consumption at each step of the life cycle, especially feed production, packaging, milk processing, and milk transportation. In contrast, extraction of mineral resources and consumption of water only require limited fuel and electricity; extraction is minor compared to direct non-renewable primary energy use. Overall impacts of fluid milk consumption on resources represent on the order of 0.3% of the annual average impact of a person living in the U.S (Table 2), whereas milk production represents 0.5% of the ecosystem impacts per U.S. person per year, mostly due to non-renewable energy use.

### 3.8. Water Footprint and Eutrophication

Water footprinting is an area of growing interest for many environmental analysts. While AWARE considers impacts of water scarcity on both humans and ecosystems, considering multiple impacts and including eutrophication impacts on freshwater and marine ecosystems is important—especially when there is important variation across the country. Figure 6 shows water stress, freshwater eutrophication, and marine eutrophication impacts per kg $FPCM_{farm}$ in each watershed. On the one hand, water scarcity is most significant in the western portion of the country, associated with both high water consumption and the high AWARE characterization factors in Western U.S. (top inset). On the other hand, freshwater eutrophication tends to be greater in the Midwest and East due to larger fate factors (bottom inset) or greater erosion. However, when restricting impacts to areas with phosphorus concentration greater than 100 μg/L, the highest freshwater eutrophication impacts are induced in the California, Missouri, and Upper Mississippi water basins (Figure 4d). Variation in marine eutrophication due to nitrogen compounds is minor relative to the other impacts, with greater impacts for the South Atlantic Gulf due to lower feed production efficiency and slightly higher characterization factors.

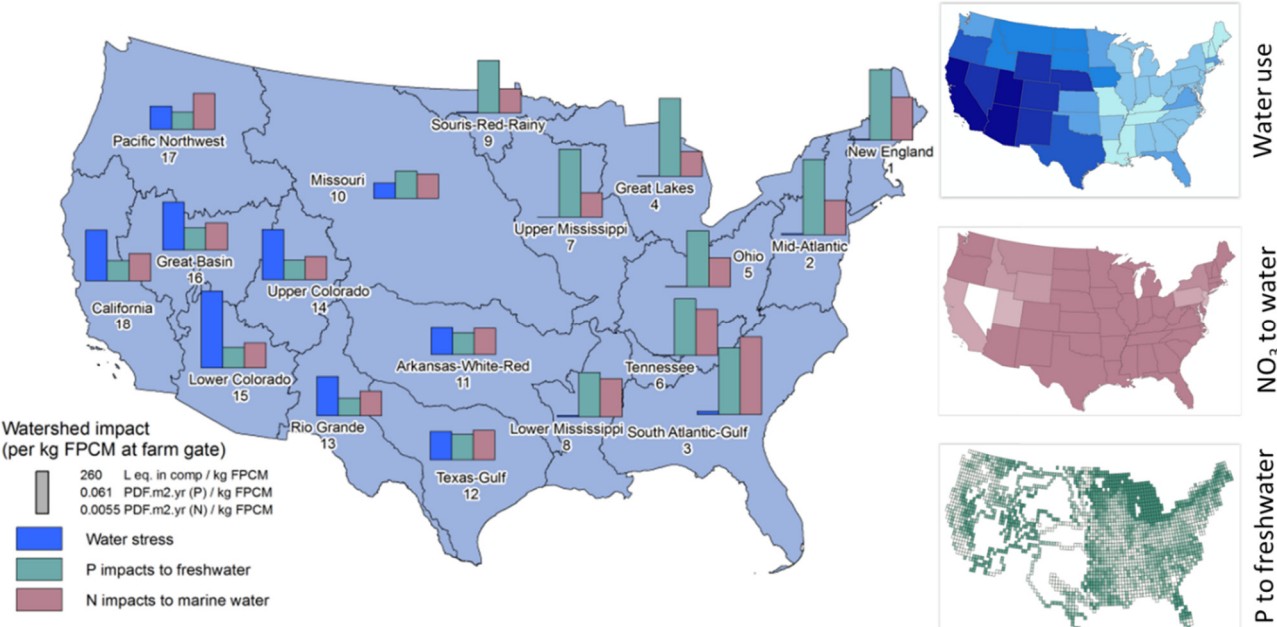

**Figure 6.** Comparison of watershed-level impacts for water stress, freshwater eutrophication, and marine eutrophication from field and dairy farm. Inset maps shows the characterization factors in native resolution applied to each watershed for; scales are not shown for visual simplicity; color gradients (light to dark) correspond to CFs values (low to high) with the following ranges: (**top**) AWARE (minimum = 0.16 to maximum = 98 $m^3_{world\ equivalents}/m^3_{in\ watershed}$), (**middle**) $NH_3$ air emissions (minimum = 0.057 to max = 0.17 PDF·m$^2$·yr/kg $NH_3$), and (**bottom**) phosphorous emissions (minimum = 0.34 to maximum = 640 PDF.m$^2$.yr/kg P). In inset maps, white areas have no CFs.

### 3.9. Comparison with Other Impact Assessment Methods and Different Products

When compared with the base assessment method, ReCiPe [64] produces largely similar results (Figure S8). Climate change impacts are also translated into dominant contributions to human health and in a lesser extent to ecosystem impacts, on-farm emissions also dominating climate change with fine particulate. For ecosystem and agricultural land use the impacts of feed production are even more dominant with ReCiPe than with the adapted version of IMPACT 2002+. TRACI [65] also largely identifies the same main processes with high contributions to on farm and manure emissions for global warming, acidification and marine eutrophication, and feed production for freshwater eutrophication and ecotoxicity (Figure S9). One main difference from IMPACT 2002+ is the dominating influence in TRACI's results of consumer use on carcinogenic and non-carcinogenic impacts, which are due to lead emissions to water associated with waste treatment of plastic milk containers.

### 3.10. Limitations and Further Needs

This study has integrated data collected, of necessity, at a variety of spatial and temporal scales. For example, the national irrigation survey is not conducted in the same years as the census of crop production. For certain impact categories and locations (e.g., water consumption and areas with high irrigation needs) it would be useful to have data available at a finer temporal and spatial resolution. For example, the state of Nebraska has relatively high corn production and relatively high water consumption impacts, making it a hot spot for corn grain water stress. However, the prevalence of irrigation varies across Nebraska, and a higher resolution could provide a more representative state value for corn grain water stress.

For eutrophication, this study relied on a spatial resolution comparable to that used by the phosphorus fate model (0.5° bx 0.5°), but state-based transport factors from TRACI were used for marine eutrophication due to nitrogen, since the variation in N emissions

from agricultural fields was found to be limited. There is also a need to further develop the effect modeling for eutrophication in LCIA, as the combined effect of phosphorus and nitrogen depend on the existing water quality of a receiving body.

For characterizing land use impacts on biodiversity, and ultimately ecosystem service potential, there is a need for a finer spatialization and distinction between a wider range of land use classes, exploring how dairy production could make more use of land not suited for direct human food production, while maintaining good feed efficiency.

For human toxicity and ecotoxicity, there is still high uncertainty, mostly related to differences in toxicity effect factors, for example associated with the inclusion or exclusion of sensitive arthropods in determining the effect factors of cyfluthrin and lambda-cyhalothrin. In addition, further research is needed in LCA on the complex question of the antibiotic resistance caused by antibiotic use during animal growth stages, since antibiotic use is already restricted for milking cows.

## 4. Conclusions

This study provides a systematic, national-scale perspective on the environmental impacts of milk production and consumption in the United States, showing geographical variation in inputs, feed and dairy farm practices, and impacts. It includes and demonstrates, in a unique way, the importance of spatialization. Although water consumption impacts have the highest spatial gradients, this study also shows spatial variability in other impacts: for eutrophication, variations in impact of at least two orders of magnitude occur between areas due to (a) geographic and crop variations, such as topography, rainfall, temperature, crop coverage, and (b) hydrologic parameters such as the distribution and prevalence of lakes and rivers, which affect the residence time of inland water. For toxicity, spatial variation of characterization factors for a given compound is more restricted, typically to an order of magnitude. In contrast, inter-chemical comparisons show 10 to 12 orders of magnitude differences in characterization factors.

The matrix approach presented in this study is a key development to structure a complex analysis that integrates local, regional, and national impacts from the cradle to farm gate, allowing for the disaggregation impacts according to location of the emissions or location of impacts. Building on this spatial matrix framework, we provide updated estimates of impacts of milk production and consumption, identifying opportunities for improvement.

This study provides important insights on the main stages contributing to impacts and on opportunities for impact mitigation: for many impact categories, feed production is dominant, emphasizing the importance of high feed efficiency, of adopting the most efficient water irrigation technologies and techniques, and of improved nutrient management to reduce fertilizer losses and costs. Direct emissions at farm are also critical for (enteric) $CH_4$ and $N_2O$ emissions, which represent an important share of the climate change impacts; in addition, barn and manure $NH_3$ emissions contribute to impacts on human health and eutrophication. Manure management is another key part of the life cycle assessment of milk production, warranting analysis and incentivization of technologies that reduce global warming impacts and nutrient losses, such as anaerobic digesters. Adequate nutrient management plans, both on farm and during milk processing, are critical to reduce nutrient losses. Reducing fine particulate matter, NOx emissions, and fuel use from tractors and milk transportation can have immediate influence on mitigating human health impacts. Decreasing electricity use throughout the life cycle represents opportunities for win-win cost and environmental benefits, with refrigeration during distribution, retail and consumption life cycle stages offering significant opportunity for electricity savings.

Finally, product loss rates are a particularly important contributor to life cycle impacts of milk production and consumption. Losses due to spoilage (during distribution, retail, and consumer stages) as well as direct waste (often by the consumer) act as multiplicative factors to all other stages of life cycle, for losses require producing more milk than is consumed. One potential avenue to reduce losses is to investigate the feasibility and benefits of producing milk products with extended shelf lives.

**Supplementary Materials:** The following supporting information can be downloaded at: https://www.mdpi.com/article/10.3390/su15031890/s1. See Supplementary Materials File.

**Author Contributions:** Conceptualization, Y.W. and O.J.; Data curation, A.D.H., A.A.-B., M.C.H. and O.J.; Formal analysis, A.D.H., A.A.-B., M.C.H. and O.J.; Funding acquisition, O.J.; Methodology, A.D.H., A.A.-B., M.C.H., J.B., D.K., L.L., M.M., R.S., M.D.M., G.T., Y.W. and O.J.; Project administration, Y.W. and O.J.; Supervision, O.J.; Visualization, A.D.H. and O.J.; Writing—original draft, A.D.H., A.A.-B., M.C.H., J.B., D.K., L.L., M.M., R.S., M.D.M., G.T., Y.W. and O.J.; Writing—review & editing, A.D.H. and O.J. All authors have read and agreed to the published version of the manuscript.

**Funding:** This research was funded by the Dairy Research Institute, Chicago, IL, USA.

**Institutional Review Board Statement:** Not applicable.

**Informed Consent Statement:** Not applicable.

**Data Availability Statement:** Matrices described in the text are available via the link shown in the Supporting Information.

**Acknowledgments:** We thank Sebastien Humbert, Emily MacDonald, and Samuel Vionnet for their scientific support.

**Conflicts of Interest:** The funding agency participated in research meetings and contributed to design of methodology and drafting of the manuscript. The funders had no role in the decision to publish the results.

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
