# Peer review of "Spatialized Life Cycle Assessment of Fluid Milk Production and Consumption in the United States"

_sustainability, doi:10.3390/su15031890_

Round 1

Reviewer 1 Report

The manuscript “Spatialized Life Cycle Assessment of Fluid Milk Production and Consumption in the United States” deals with the inclusion of geographical differencens in the LCA-based evaluation of U.S. milk production and consumption.

The manuscript is well written, although a bit complex since a huge amount of results has been reported. Therefore, I suggest simplyfing the text, where possible.

Some minor comments:

line 225. I think that some words lack in this sentences. Please check

line 237. The unit Leq in competition is not clear stand-alone. Please include also the impact category to which it is associated with

line 347. "are"? Please check the sentence

line 399 "ihpa8cl"?

line 400. Please do not star a sentence with acronyms. Check throughout the manuscript.

line 435. Check the value of GTP, since it is different with respect to that reported in table 1

Figure 2. The cross-hatching associated with "retail" ad "consumer" is not clearly linked to the milk losses in the caption (in the main text is undestandable). Please complete the caption stating clearly that the relationship between the cross-hatching and the losses

Reviewer 2 Report

This paper documents a very comprehensive farmgate and full LCA of milk consumption in the U.S. Although it is not possible to verify all the results provided, all values seem reasonable. The work is well done and relatively well documented. My comments and suggestions are relatively minor.

1.       In the abstract and in the main text, it should be stated that much of the data used in the LCA was from around 2007; at least that seems to be the case for the farm data. Changes in production have occurred since then which would affect the results. The general results, comparisons and conclusions would not be affected much with this older data, but it is important to document the time when the data were collected.

2.       It is not clear how you differentiate between home-grown feeds and those purchased elsewhere. It seems they should be treated differently, particularly for water consumption and water quality.

3.       Many places through the manuscript the term “higher” is used where “greater” would be more appropriate. Consider doing a search and replace to modify many of these.

Specific lines:

Line 47- This sentence is unclear on what impacts you are referring too. Is this all impact categories assessed?

Line 117 – I don’t think this is exactly true. Most feed was produced on the farm where irrigation was used as required.

Line 185-188: It is not clear how this connection was done.

Figure 1: This figure should stand alone; therefore, all abbreviations/parameters should be defined. Not clear how i and j are different. Are areas k and l different? “feed produced” should be defined as “dairy feed produced” or are you modeling all feed produced and allocating a portion to dairy. Milk cans are very old technology, so this may not be the best picture to use here.

Line 225: Something is wrong here, “in in 17”. Is this referring to supplementary information?

Line 252: delete “of”

Equation 3: I don’t think “Inational milk consumed” is defined anywhere.

Line 267: This sentence is unclear as written. Reference Thoma’s paper. Are these SI tables published by Thoma?

Line 306: …were systematically…

Lines 308-309: This could benefit from a little more explanation.

Line 347: delete “are”

Line 382: Now that the AR6 GWP values are released, you may want to comment on those, which would be much more like the AR4 values. You could consider replacing the AR5 values with AR6 values.

Line 396: …emission flows…

Line 409: FPCM

Line 429: …greenhouse gas emission flows…

Figure 2: It is not clear how you are indicating consumer losses and allocation to cream. They look the same in the figure.

Figures 3&4: The figure should stand alone so all abbreviations should be defined, especially the y axes.

Lines 585-587: These acre values should probably be converted to hectares.

Table 2: define DALY and PDF.

Figure 6: Is there a scale for these figures, particularly those on the right. Are these just very general representations? I assume darker means more and white means little or none.

Reviewer 3 Report

1-In order to improve the scope of your work I suggest that you summarize the main results in terms of achieving the initial objectives (compared with) of the work (three objectives).   [ This comparison must lead to a list of recommendations, both in the perspective of a revision of the product design and for the optimization of its use in terms of environmental impacts, (we find some in your manuscript recommendations but scattered in the text).]

2-It will be very interesting if you dedicate a paragraph that clearly exposes the results of the life cycle analysis in which you quantify the strengths and weaknesses of the products, to identify the determining parameters in terms of environmental impacts. All of these practices help to improve the process of improving the ecological quality of products.

Author Response

Overall response: We thank the reviewer for their time and comments.

1-In order to improve the scope of your work I suggest that you summarize the main results in terms of achieving the initial objectives (compared with) of the work (three objectives).   [ This comparison must lead to a list of recommendations, both in the perspective of a revision of the product design and for the optimization of its use in terms of environmental impacts, (we find some in your manuscript recommendations but scattered in the text).]

Response:  We appreciate the need to link results back to objectives, and this was not adequately clear.  We have added some sentences to the conclusions to provide that link; we believe that the conclusions also include the list of recommendations.

2-It will be very interesting if you dedicate a paragraph that clearly exposes the results of the life cycle analysis in which you quantify the strengths and weaknesses of the products, to identify the determining parameters in terms of environmental impacts. All of these practices help to improve the process of improving the ecological quality of products.

Response: Since we have written this manuscript with the scientific community in mind, our focus has been on the method required to adequately characterize the impacts of a complex, spatially heterogeneous product. While we have not emphasized the strengths of milk, we hope that the conclusions do provide adequate areas of focus for improving milk production and consumption.